# The diversity and habitat association of medium and large mammals in the Dhidhessa Wildlife Sanctuary, Southwestern Ethiopia

**Girma Gizachew Tefera**[ORCID][1]*, **Tadesse Habtamu Tessema**[2], **Tibebu Alemu Bekere**[1], **Tariku Mekonnen Gutema**[1]

**1** Department of Natural Resource Management, Colleague of Agriculture and Veterinary Medicine, Jimma University, Jimma, Ethiopia, **2** Department of Biology, College of computational Sciences, Jimma University, Jimma, Ethiopia

* kennaa20047@gmail.com

## Abstract

Understanding species diversity and habitat association is the baseline for developing conservation plan. The study aimed to assess diversity, abundance and habitat association of medium and large sized mammals in the Dhidhessa Wildlife Sanctuary (DWS), Southwestern Ethiopia. The survey was conducted from December 2022 to July 2023, both in the wet and dry seasons. A stratified random sampling design was applied to stratify the study area in to four (wooded grassland, riparian forest, seasonally flooded grassland, and savanna grassland) strata based on vegetation and habitat type. Lines transect survey, sensor camera trapping, and indirect and direct evidence methods were used to collect data during wet and dry seasons. Data were analyzed using the chi square test and species diversity indexes. Twenty -seven mammalian species were recorded for the area. Order Artiodactyla which had the highest number of species (eleven = 11) followed by the, order Carnivora (n = 9). While, orders Rodentia and Tubulidentata each represented by one species. Papio anubis (n = 500, 24.9%) were the most abundant species followed by Hippopotamus amphibious (n = 364, 17.8%) in the present study area in both wet and dry seasons. But Panthera pardus (n = 13, 0.64%) and Civettictis civetta (n = 13, 0.64%) were the least abundant species in the present study area. Riparian forest had the highest number of species (n = 732, 36.3%) followed by savanna grassland (n = 615, 30.5%). Savanna grassland held the highest species diversity of medium- and large-sized mammals (H′ = 2.44). Seasonally flooded grasslands (E = 0.6849) and riparian forests (E = 0.4889) showed the highest and lowest evenness of the mammalian species, respectively. Therefore, DWS's primary priority should be creating management plans to reestablish the sanctuary as a fully functional sustainable ecosystem and ensuring the social and economic viability of the surrounding community.

**Data availability statement:** Data relevant to this paper are available from figshare at https://figshare.com/s/714b79120abd3a2ef204.

**Funding:** The author(s) received no specific funding for this work.

**Competing interests:** The authors declare no conflict of interest.

# Introduction

## Background of the study

Mammals are highly versatile group due to their diversity in species, morphologies, ecologies, and behaviors [1]. The class Mammalia contains 5487 species globally. On the African continent, there are more than 1150 distinct mammalian species [2,3], with 360 of those species found in eastern Africa [2], and 320 in Ethiopia [4,5]. Fifty-seven of the 326 mammalian species known to exist in Ethiopia are endemic [5–7]. The weight of a medium-sized mammal ranged from 2 to 7 kg, whereas that of a large mammal was greater than 7 kg and medium sized, with large mammals accounting for more than 60% of the total [4]. The existence of several endemic species has been facilitated by changes in the temperature, geography, and vegetation. There are many species of mammals and other higher vertebrates in Ethiopia, which is reflected in the country's great faunal richness [4,6,7]. Mammals can be found in almost all habitats and place on the earth except ice of Antarctica from an elevation below sea level up to 6500 m above sea level [7]. Terrestrial habitats are home to the vast majority of mammalian species (98%) [8]. In Ethiopia, medium and large sized mammals are widely distributed in a wide range of ecosystem from desert to afro- alpine ecosystem [9]. However, the characteristics of the habitats are thought to be important determinants of the number and variety of mammal populations [8]. Food and shelter availability, habitat stability and heterogeneity, seasonal climate variation, predator status, and other ecological needs like the type and density of vegetation in their habitat all affect mammal distribution and abundance [3].

In varying seasons and habitat types, mammals are not uniformly dispersed [9]. The main cause of this is that each species has a unique range that is the outcome of the interaction between the ecological conditions of the present day and the species' evolutionary history. However, many species have similar, though not always identical, distribution patterns. Conversely, individual distribution within a region of easily accessible habitats is indicated by home ranges, territories, and microhabitats [8]. As to [7], medium- and large-sized mammals have heightened sensitivity to alterations in their habitat, rendering them dependable markers of the environmental condition. Structurally complex ecosystems may provide more niches and diverse methods of utilizing natural resources, increasing species diversity [8,9].

Mammals are essential to the healthy structure and operation of ecosystems [10,11]. They have the ability to change the structure of vegetation, alter pathways of nutrients, and ultimately, change species composition of the ecosystem [11]. As significant seed dispersers for numerous tree species, they participate at various trophic levels in food webs and contributing to herbivore regulation [10,11]. Despite this, mammals provide humans with a number of economic advantages, such as food, pets, antiques, and tourism [12,13]. However, recently escalating natural and anthropogenic factors greatly threatened survival of mammals [13,14]. The main natural and human-induced activities that caused the depletion of mammal species include habitat loss and fragmentation, overexploitation, global climate change and other anthropogenic pressures threaten mammals worldwide [8,14]. A recent assessment of the conservation status of mammal species indicated that at least one-fifth of mammal species are at risk of extinction in the wild worldwide [8,15]. According to [15] a study on the prehistoric distribution of 173 mammals across six continents showed that, half of their range area has disappeared. Similarly, for a century, illegally mammal hunting for meat and other uses has continued even in the national parks in developing countries [16–18]. As a result, 40% of the mammal species that are considered Critically Endangered are impacted by hunting [8,18,19]. In developing countries such as Ethiopia, rapid population growth and the resulting increase in demand for agricultural land to increase productivity are predicted to pose a serious threat to biodiversity [14,20,21].

In Ethiopia, mammals in particular and wildlife in general are somewhat conserved in protected areas including national parks, sanctuaries, controlled hunting areas, Biosphere, community conservation areas, and others [5,22,23]. Despite an increase in the number of mammal taxa recorded for Ethiopia, there is still no comprehensive inventory or thorough documenting of the species of mammals found in Ethiopia's many habitats [4]. According to [16,17], data on the status and trends of mammals is necessary for adequate management and protection of protected areas.

The diversity of wildlife species found in the DWS is not supported by scientific evidence or official documentation, but numerous oral histories and reports from local informants attest to the abundance of wildlife, particularly mega herbivores and large carnivores, and support the sanctuary's designation from fifty years ago. The country's wildlife resources, including those found in well-established protected areas, have been severely depleted over the past thirty years for a variety of reasons. These include civil war, widespread settlement in the then-wilderness areas, such as the Dhidhesa wildlife sanctuary, widespread settlement during periods of drought, and the unstable and quickly changing central government [7,24]. Because of the extensive anthropogenic activity around the DWS, a greater portion of the wildlife habitats have been destroyed far earlier than the wildlife resources in the area have been documented. This poses a threat to wildlife habitats. In addition, the Arjo Dhidhessa Sugar Factory is the result of the conversion of nearby regions surrounding the sanctuary into automated sugarcane farms [24]. Therefore, in order to provide baseline data for efficient wild animal conservation, the study intends to assess and document the species diversity and habitat association of medium- and large-sized mammals in the DWS.

## The study area and methods

### Ethical statement

The Review Board at Jimma University (JU) College of Agriculture, Veterinary, and Medicine evaluated and approved the research proposal. The vice president's office also reaffirmed and endorsed the review board's request for research and community service. JU permitted the necessary procedures for obtaining consent related to the data collection tools (RGS/752/2021). After receiving approval from the human ethics research committee, we obtained an authorization letter and a request for collaboration from all Kebeles and villages to conduct this research project in the Dhidhessa Wildlife Sanctuary and its surrounding area southwestern Ethiopia.

### Description of the study area

The research was carried out in the DWS area and its vicinity in southwestern Ethiopia. This region is situated 395 kilometers from Addis Ababa, positioned between 8°36'0" and 8°48'0" N latitude and 36°21'0" and 36°33'0" E longitude, covering an area of about 1300 square kilometers and shared among three administrative zones: East Wollega, Buno Bedele, and Jimma, all within the Oromia National Regional State (Fig 1).

The altitudes in the DWS vary between 1350 and 1050 meters above sea level. This region experiences two distinct seasons: a dry (November to February) and a wet (June to September), with precipitation levels ranging from 648 mm to 2001.8 mm. The DWS has relatively warm temperatures, with a mean annual minimum of 12°C and a maximum of 35°C [25]. Besides the extensive savanna and wetlands along the riverbanks, the area's terrain features moderately rugged, hilly, and mountainous landscapes. The habitats in the study area primarily consist of natural forests, wooded areas, riparian forests, seasonally flooded prairies, and broad savannah prairies, which are formed by river overflows and their permanent

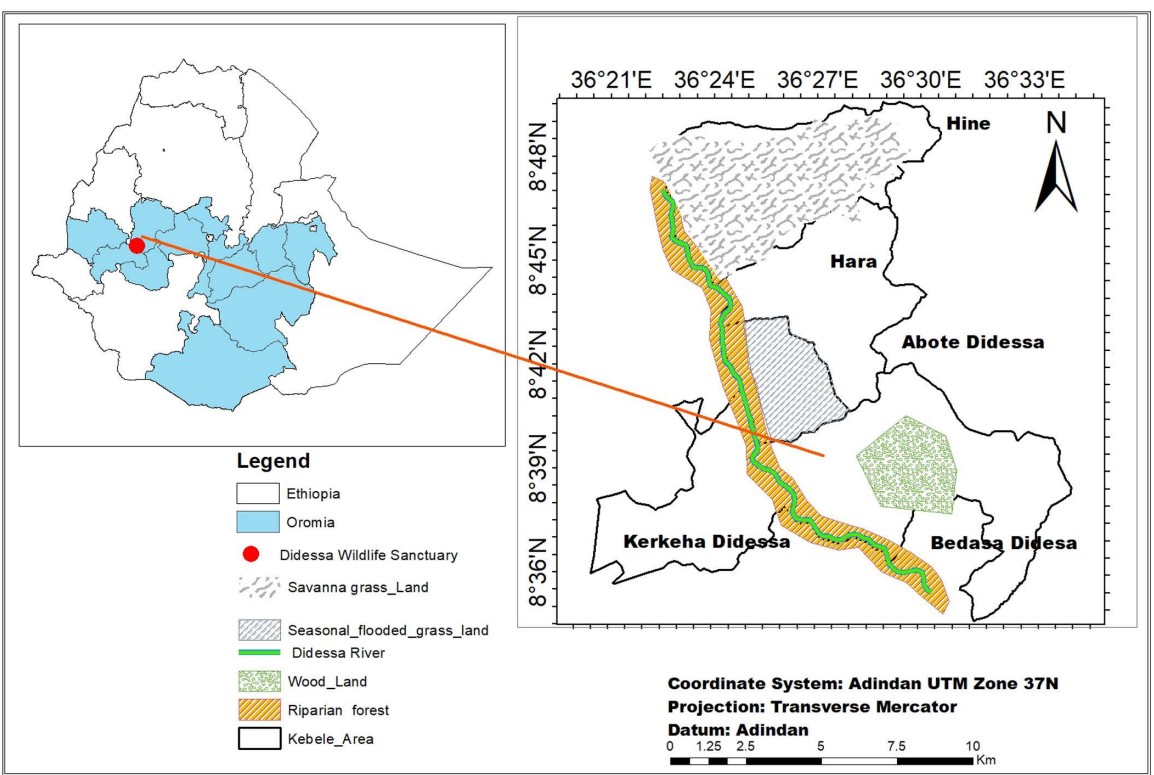

**Fig 1. Map of study area (Our team used shape files provided by the Ethiopian Mapping Agency to produce the map.** The agency's official website offers free access to these shape files (http://www.ema.gov.et).

tributaries. The DWS was formerly designed by the Ethiopian government in 1970 with the objective of conserving a variety of wildlife, including endangered mammals like African wild dog (*L. pictus*) and African elephant (*L. africana*). Recently, part of its area was designated as a nominally controlled hunting area; the Haro Abba Dikko controlled hunting area, with a special focus on conservation of African buffalo. Subsistence farming and large-scale sugar-cane farms, however, are practiced throughout the human-dominated remaining portions of the DWS. Subsequently, L. pictus and L. africana had extirpated from the DWS. Despite the loss of wildlife habitats in the area, large mammals such as African buffalo (Syncerus caffer), lions (Panthera leo), leopards (Panthera paradus), hippopotamus (Hippopotamus amphibius), many more medium- and large-sized mammals and nonhuman primates are still found in the remnant forest patches of the DWS. Furthermore, the research area's vegetation is primarily composed of *Combretum molle*, *Terminalia brownii*, *Piliostigma thonningii*, *Cordia Africana*, *Entada abyssinica*, *Acacia sp.*, *Ficus sp.*, *Fagaropsis angolensis*, with *Hyparrhenia* savanna grass being the most prevalent. As a result, some evaluative reports show that the DWS prematurely hindered itself from achieving the objectives for which it was established [24].

## Methods

A preliminary survey was carried out in February 2021. The DWS was characterized by heterogeneous habitats. Based on the vegetation types, the habitat types of the study area were classified into four (riparian forest, savanna grassland, seasonally flooded grassland and wooded grassland) [2,26]. During the preliminary survey, the physical features of the study area were assessed using ground survey and the coordinates of each study site was taken

and their boundaries was defined. The sampling unit within the study area determined and assigned on the basis of size of the habitat (Table 1).

Transect lines counts were used to determine the diversity, and habitat association of medium and large sized mammals. Transect lines was placed on the stratified habitat types in a random fashion and proportional to an area of the habitat type [27]. A total of 25 lines transect, covering about 25% (24.65 km²) of the study area. Wildlife trails, footpaths, and other access routes were also used as transect lines. The average length of line transect was five kilo meters and the width were determined by the vegetation type and season, and ranges between 75 meters (riverine and tall grass bearing wooded grassland) and 200 m (open grasslands) and each transect ends about a km from the habitat edge [28]. Adjacent transect lines were spaced between one and two kilo meters to avoid double counting.

Data collection were conducted twice a day for three days in each month during both the dry seasons (December, January and February) and the wet (May, June, and July) seasons, from 06:00 to 10:00 in the morning and 16:00 to 18:00 in the late afternoon, when the animals were active and visibility was good [29,30]. The species of mammals were identified using field guidebooks and locals people. Simultaneous counting and thorough observations of animal eating and sleeping areas, particularly on the cliff, were used to prevent counting the same species or individual animals again. Upon detection of animals; habitat types, season of the year, and number of individuals observed was record. To acquire correct data, well-experienced researchers with wild animal expertise were involved [31,32].

During the line transect survey, observations were made with the Bushnell wildlife camera, traps naked eye, or aided by binoculars (7 x 50 mm) while evidence of tracks, scats, dung, dens, burrows, carcasses, feeding remains, scratches, feeds, beds, and calls were considered as indirect observations. Indirect pieces of evidence are very useful when surveying animals that are naturally rare, elusive, and found at low densities [33].

## Data analysis

Collected data were organized and entered into a worksheet for the analysis. All statistical analyses were performed using the PAST 4 software. PAST 4 was used to analyze the species diversity of mammals, Species evenness, and Simpson similarity index. The relative abundance index of species (RAI) was calculated by dividing the number of records of each species by the total number of records of all species. The abundance of observed mammals was categorized as common if they were seen during all of the surveys, uncommon if they were seen in more than half of the surveys, and rare if seen less than half of the surveys. A Chi-square ($\chi^2$) test of independence was used to examine differences in the abundance of medium and large-sized mammal species among habitats and between the wet and dry seasons [34]. Moreover, the conservation status of mammals identified from DWS was described as vulnerable and least concerned as per the IUCN Red List [35]. Additional information for the presence of mammal species was supported by focus group discussion with selected local community

Table 1.  Area of study and transect line.

| Habitat type | Area (km²) | Actual transects | Number of sample transects | Length and width of the transects (km) |
|---|---|---|---|---|
| Riparian forest | 17.75 | 18 | 4 | 5x0.2 |
| Wooded grassland | 38.5 | 39 | 10 | 5x0.2 |
| Savanna grassland | 27.2 | 27 | 7 | 5x0.2 |
| Seasonally flooded grass land | 15.11 | 15 | 4 | 5x0.2 |
| Total | 98.56 | | 25 | |

using field guides. The presence of potential anthropogenic wildlife threats was also recorded during field observation.

## Results

### Species composition and richness

Twenty-seven species of medium- and large-sized animals from five orders and twelve families were found in the study area in both dry and rainy seasons (Table 2. Five medium species—the crested porcupine (*Histrix cristata*), honey badger (*Mellivora capensis*), black-blacked jackal (*Canis mesomelas*), common jackal (*Canis aureus*), and aardvark (*Orycteropus afer*)—accounted for 18.5% of the total. The remaining 22 species, or 81.5%, were large mammals) (S1 Fig in S1 Appendix).

With eleven species, the order Artiodactyla has the most of all the orders, followed by the order Carnivora with nine species. Conversely, one species each represented the orders Rodentia and Tubulidentata. With seven species, the Bovidae family was the largest family, followed by the Cercopithecidae family. On the other hand, only one species represented each of the six families.

**Table 2. Mammalian species identified in study area (F = Foot prints; S = Seen; P = Photographs; D = Droppings; B = Borrows, \* = Community information).**

| Order | Family | Species | Common name | Local name (Oromifa) | Sign recorded |
|---|---|---|---|---|---|
| Primate | Cercopithecidae | *Cercopithecus mittis* | Blue monkey | chano | F, S, P |
| | | *Chlorcebus aethiops* | Vervet monkey | Qamalee | S, P, F |
| | | *Papio anubis* | Anubis baboon | Jaldeesa | S, F, P |
| | | *Cercopethicus neglectus* | DeBrazzes monkey | – | S, P |
| | Colobinae | *Colobus guerza* | Gureza | weenni | S, P, F |
| Artiodactyla | Suidae | *Phacochoerus africanus* | Warthog | Karkaroo | P, S, D |
| | | *Potamochaerus larvatus* | Bush pig | Booyye | S, P |
| | | *Hylochoerus meinertzhageni* | Giant forest hog | Booyye | P |
| | Hippopotamidae | *Hippopotamus amphibius* | Hippopotamus | Roobii | S, P, F |
| | Bovidae | *Syncerus caffer* | African buffalo | Gafarsa | S, P, F, D |
| | | *Tragelaphus scriptus* | Bush buck | Bosonu | P, S |
| | | *Sylvicapra grimmia* | Bush Duiker | Quruphe | P, F, D |
| | | *Kobus ellipsiprymnus* | water buck | Worabbaa | \* |
| | | *Oreotragus oreotragus* | Klipspringer | | \* |
| | | *Redunca redunca* | Reedbuck | | \* |
| | | *Ourebia ourebi* | Oribi | | \* |
| Carnivora | Felidae | *Panthera leo* | Lion | Leenca | F, D, \* |
| | | *Panthera pardus* | Leopard | Qerrensa | P, D |
| | | *Leptailurus serval* | Serval cat | Iya | \* |
| | | *Felis caracal* | Caracal | | \* |
| | Hynaenidae | *Crocuta Crocuta* | Spotted Hyena | Warabessa | P, D |
| | Mustelidae | *Mellivora capensis* | Honey badger | Amaaqexa | \* |
| | Viverridae | *Civettictis civetta* | African civet | Xirinyi | \* |
| | Canidae | *Canis aureus* | Common jackal | Wangoo | F, P |
| | | *Canis mesomelas* | Black blacked jackal | Jeedala | \* |
| Rodentia | Hystricidae | *Histrix cristata* | Crested porcupine | Dhaddee | P, S |
| Tubulidentata | Orycteropodidae | *Orycteropus afer* | Aardvark | Waldigessa | S |

## Species relative abundance

There were significant variations in the seasonal species abundance of medium- and large-sized mammals, with a total of 2019 individuals reported, of which 1093 (54.1%) were observed during the dry season and 926 (45.9%) individuals during the rainy season ($\chi2 = 13.81$, df = 1, p < 0.05). *Papio anubis* represented (n = 500, 24.9%) was the most abundant species individuals recorded followed by *Hippopotamus amphibious* (n = 364, 17.8%). On the other hand, *Civettictis civetta* (n = 13, 0.64%) and *Panthera pardus* (n = 13, 0.64%) were the two least abundant species (Table 3).

Out of the 27 species of mammals identified in DWS, 11 (40.7%) were considered uncommon, 9 (33.3%) common, and 7 (25.9%) rare. Comparably, Table 4 demonstrates that of the known mammalian species, about 70.4% were categorized as least concern and 14.8% as vulnerable. However, the remaining 14.8% are thought to be conservation-dependent, near threatened, and at lower risk (Table 4).

**Table 3. Relative abundances of mammalian species recorded in the study areas.**

| Species name | Number of individuals recorded | | Relative abundance (%) | | |
|---|---|---|---|---|---|
| | Dry season | Wet season | Dry season | Wet season | Total |
| *Cercopithecus mittis* | 75 | 53 | 6.9 | 5.7 | 6.3 |
| *Chlorocebus aethiops* | 142 | 127 | 12.9 | 13.7 | 13.4 |
| *Papio anubis* | 261 | 239 | 23.9 | 25.8 | 24.9 |
| *Cercopethicus neglectus* | 28 | 17 | 2.56 | 1.84 | 2.2 |
| *Colobus guerza* | 87 | 79 | 7.96 | 8.5 | 8.23 |
| *Phacochoerus africanus* | 13 | 8 | 1.19 | 0.9 | 1.05 |
| *Hylochoerus meinertzhageni* | 18 | 13 | 1.7 | 1.4 | 1.6 |
| *Potamochaerus larvatus* | 37 | 24 | 3.4 | 2.6 | 3 |
| *Hippopotamus amphibius* | 231 | 133 | 21.13 | 14.4 | 17.8 |
| *Syncerus caffer* | 28 | 45 | 2.56 | 4.9 | 3.7 |
| *Tragelaphus scriptus* | 32 | 24 | 2.93 | 2.6 | 2.8 |
| *Sylvicapra grimmia* | 23 | 12 | 2.1 | 1.3 | 1.7 |
| *Kobus ellipsiprymnus* | 12 | 6 | 1.09 | 0.65 | 0.9 |
| *Oreotragus oreotragus* | * | * | * | * | * |
| *Ourebia ourebi* | * | * | * | * | * |
| *Redunca redunca* | * | * | * | * | * |
| *Panthera leo* | 12 | 9 | 1.09 | 0.97 | 1.03 |
| *Panthera pardus* | 8 | 5 | 0.73 | 0.54 | 0.64 |
| *Leptailurus serval* | 16 | 27 | 1.5 | 2.9 | 2.2 |
| *Crocuta crocuta* | 11 | 36 | 1.01 | 3.9 | 2.4 |
| *Felis caracal* | * | * | * | * | * |
| *Mellivora capensis* | 9 | 14 | 0.8 | 1.5 | 1.2 |
| *Civettictis civetta* | 8 | 5 | 0.73 | 0.54 | 0.64 |
| Canis aureus | 12 | 10 | 1.09 | 1.08 | 1.08 |
| Canis mesomelas | * | * | * | * | * |
| Histrix cristata | 19 | 17 | 1.74 | 1.84 | 1.8 |
| Orycteropus afer | 11 | 23 | 1 | 2.48 | 1.74 |
| Total | 1093 | 926 | 100 | 100 | 100 |

**Table 4. Occurrence of medium and large sized mammals in the study areas.**

| Species name | Local name | Occurrence category | IUCN category |
|---|---|---|---|
| *Cercopithecus mittis* | Blue monkey | Common | VU |
| *Chlorocebus aethiops* | Vervet monkey | Common | LC |
| *Papio Anubis* | Anubis baboon | Common | LC |
| *Cercopethicus neglectus* | DeBrazzes monkey | Common | LC |
| *Colobus guerza* | Gureza | Common | LC |
| *Phacochoerus africanus* | Warthog | Uncommon | LR/LC |
| *Hylochoerus meinertzhageni* | Bush pig | Common | LR/LC |
| *Potamochaerus larvatus* | Giant forest hog | Common | LR/LC |
| *Hippopotamus amphibius* | Hippopotamus | Common | VU |
| *Syncerus caffer* | African buffalo | Uncommon | NT |
| *Tragelaphus scriptus* | Bush buck | Uncommon | LC |
| *Sylvicapra grimmia* | Bush Duiker | Uncommon | LR/LC |
| *Kobus ellipsiprymnus* | water buck | Uncommon | LC |
| *Oreotragus oreotragus* | Klipspringer | Rare | LC |
| *Redunca redunca* | Reedbuck | Rare | LR/cd |
| *Ourebia ourebi* | Oribi | Rare | LR/cd |
| *Panthera leo* | Lion | Uncommon | VU |
| *Panthera pardus* | Leopard | Rare | VU |
| *Leptailurus serval* | Serval cat | Rare | LC |
| *Crocuta crocuta* | Spotted Hyena | Common | LC |
| *Hyaena hyaena* | Striped hyena | Rare | NT |
| *Mellivora capensis* | Honey badger | Uncommon | LC |
| *Civettictis civetta* | African civet | Uncommon | LC |
| *Canis aureus* | Common jackal | Uncommon | LC |
| *Canis mesomelas* | Black blacked jackal | Rare | LC |
| *Histrix cristata* | Crested porcupine | Uncommon | LC |
| *Orycteropus afer* | Aardvark | Uncommon | LC |

Abbreviations: LR/cd, Lower Risk/ conservation dependent; LR/LC, Lower Risk/ least concern; LC, Least Concern; NT, Near Threatened; VU, Vulnerable.

## Habitat association of mammals

In the research area, seasonal variations in the distribution of medium and large mammal species were noted (Table 5). For example, mammals regularly used riparian forests and savanna grasslands in both wet and dry seasons. In contrast, animals used the seasonal flooded grassland (9.3%) less frequently in both seasons (Table 5). There was a noticeable difference in the distribution of indicators linked to mammals' observations in various habitat types during the wet and dry seasons ($\chi2 = 18.87$, df = 3, $p < .05$).

## Species diversity indices

Table 6 shows that compared to other habitats, the Shannon diversity of mammal species was higher in the savanna grassland (H = 2.44) and wooded land (H = 2.35). Conversely, seasonal flooded grasslands had the lowest species diversity (H′ = 2.18). Nonetheless, there was no significant variation in the Shannon-Wiener Index values among the four habitat types. Seasonally flooded grasslands (E = 0.6849) and riparian forests (E = 0.4889) showed the highest and lowest evenness of the mammalian species, respectively. On the

**Table 5.  Habitat association (%) of large mammals during the wet and dry seasons in the study area.**

| Season | Number of individuals observed in different habitats (%) | | | |
|---|---|---|---|---|
| | Riparian forest | Wooded land | Savanna grassland | Seasonal flooded grassland |
| Dry | 393 (36) | 232 (21.2) | 328 (30) | 140(12.8) |
| Wet | 339 (36.6) | 252 (27.2) | 287 (31) | 48 (5.18) |
| Total | 732 (36.3) | 484 (23.97) | 615 (30.5) | 188 (9.3) |

**Table 6.  Measuring of biodiversity (Index of diversity) in DWS.**

| Habitats | | | | |
|---|---|---|---|---|
| Variables | Riparian forest | Wooded land | Savanna grassland | Seasonal flooded grassland |
| No. of species | 20 | 19 | 19 | 13 |
| No. individuals | 732 | 484 | 615 | 188 |
| Dominance_D | 0.1482 | 0.1339 | 0.1239 | 0.147 |
| Simpson_1-D | 0.8518 | 0.8661 | 0.8761 | 0.853 |
| Shannon_H | 2.28 | 2.346 | 2.44 | 2.186 |
| Evenness | 0.4889 | 0.5496 | 0.6036 | 0.6849 |

other hand, in the riparian forest (D = 0.1482) and the savanna grassland (D = 0.1239), the highest and lowest dominance of mammalian species were observed (S1 Table in S1 Appendix). Additionally, during the dry and wet seasons, the dominance of mammalian species and the Shannon different characteristics record were comparable (S2 Table in S1 Appendix). The research area's maximum species diversity (0.8761) was shown by Simpson's index of diversity.

## Discussion

### Species composition and richness

The areas were classified into four habitat types (riparian forest, savanna grassland, seasonally flooded grassland and wooded grassland) based on land covering. The study identified 22 large and 5 medium-sized animal species from five orders and twelve families. In compared to previous studies, there were comparatively high species. Mammal studies with similar goals and methodologies to this one have been conducted both outside the country and in different parts of Ethiopia, but the number of mammals in DWS was substantial. For example, the 27 species of medium- and large-sized mammals found in the current study area had a significantly higher diversity than the 23 species of mammals found in Arawale National Reserve, Kenya [36], the 23 species found in the Baroye control hunting area [37], the 22 species found in the fragmented remnant forest near Asella town [18], a total of 18 species were observed by [2], around the Wondo Genet forest area, 14 species of medium- and large-sized mammals were recorded in Yayu forest by [38], and 12 species of mammals were identified in the Wabe forest's fragments by [29]. A higher level of mammalian diversity in the current research area could have been caused by sufficient vegetation cover and resource availability. Direct observations were also used to identify the majority of the animal species found in the current research area [23] revealed that the Watch protected forest produced the same results. The fact that the habitats are so open, maybe as a result of habitat loss and fragmentation, allows for the detection of mammals with the unaided eye. The current study region has a high degree of secretiveness, illusiveness, shyness, and human disturbance, which contributes to the low number of large mammal species that were identified by indirect evidence rather than direct

sighting. This outcome was comparable to research conducted by [2] in Loka Abaya National Park and by [29] from Wabe forest fragments.

In contrast to the high diversity of animals [41] documented in Gambella National Park, Ethiopia [39] and 31 species recorded from Dhati Wolel National Park, Ethiopia [40] the results showed comparatively lesser diversity in the study area. The lower diversity of mammals observed in the current study area could be attributed to fragmentation caused by many anthropogenic activities, severe habitat loss, conflict between humans and wildlife, or low level of attitude among local residents toward conservations mammals [24]. Similarly, [41] noted that the demand for wide areas, overexploitation, and habitat fragmentation have led to decline in medium- and large-sized mammals.

The orders Artiodactyla (11 species) and Carnivora (9 species) were the most abundantly seen mammal groups in the current study region. The findings of [40] in Dhati Wolel National Park are consistent with the high number of Artiodactyla in the current research region. With the largest diversity of ungulates on Earth, the African savanna environment has supported multispecies animal production systems for thousands of years [42]. The large proportion of riverine habitats and periodically flooded grasslands that supplied enough grass for grazing during the dry season was probably responsible for the high diversity of herbivores in the current study area. In addition, during the dry season, fires occur in the savanna grassland that is dotted with Combertum-Terminalia woods. This would enable the high-nutrient new grasses to regenerate, supporting a wide variety of grazers throughout the wet season. Rainfall, fire and herbivory are the prime driving variables in African savanna ecosystems [42]. On the other hand, a large number of different and prolific herbivore species are thought to be responsible for the high diversity of carnivores seen in DWS. Large groups of herbivores, including hippopotamus and African buffalo, were regularly seen throughout the current study, suggesting their high abundance in the region and their ability to serve as a plentiful source of prey for a variety of large carnivore species [43].

## Relative abundance

In the current study area, during both the wet and dry seasons, *Papio anubis* was the species with comparatively higher abundance. Nevertheless, the two least common species were *Civettictis civetta* and *Panthera pardus*. This might be attributed to the adaptation of the species to feed on variety of food items. Similar finding was reported by [41] in Birbir protected forest western Ethiopia. In contrast, carnivores were the least common in the study area in terms of both number and distribution. Respondents claim that human-caused habitat destruction and illegal hunting have an impact on mammals' basic needs, which in turn has an impact on mammalian diversity. Additionally, they claim that the prevalence of carnivores (particularly spotted hyena and common jackal) preying on livestock in and around the research area drastically reduces the number of these animals. Similarly, least abundance of carnivores also reported in Dati Wolel National park [7].

Despite the fact that, the area is surrounded by a landscape that is dominated by humans and frequently poses a threat to their survival, many mammals have the ability to transition from diurnal to nocturnal or crepuscular activity. In the riparine forest, however, nonhuman monkey species were frequently sighted. This could be because nonhuman primates have successfully adapted to human disturbance and changing environments. Similar finding was reported by [44] in the Jorgo-Wato Protected Forest in Western Ethiopia.

## Habitat association of mammals

Mammal abundance was also influenced by habitat type, forest cover, and landscape [16]. In addition, habitat usage patterns have a significant impact on the interactions between wildlife

species and ecological communities, as well as the long-term survival and population stability of species [16,23,45]. In the research area, riparian forest supported a large number of mammal species during both seasons, followed by savanna grassland. Seasonally flooded grasslands sustained the least amount of diversity. The abundance of food, water, and vegetation cover may be the cause of this high observation rate, but the lack of refuges and vegetation cover may be the reason for the low number of mammal individuals. Similar result was reported by [16,23,41].

## Species diversity indices

The richness of species in the various ecosystems varies, as indicated by the species diversity index. The savanna grassland in the current study region, for instance, has the highest species diversity (H′ = 2.44), while the seasonal flooded grassland has the lowest species diversity (H′ = 2.18). Likewise, studies by [18] showed that variables like habitat quality, species preferences, availability of food resources, dense forest cover, and water are related to variations in the distribution and abundance of mammal species within different habitats, and that these variables are likely to lead to higher species richness. Our findings go counter to those of the study by [23,46], which indicated that the natural forest had a greater species variety. Human pressure on DWS is one potential source of these variations [24].

## Conclusion and implications for wildlife conservation

The current study provided baseline data regarding the presence and abundance of medium- and large-sized animal species in the DWS, southwestern Ethiopia, as well as information about numerous anthropogenic activities. DWS is unique in that it is home to animals with great conservation value, including large mammals like *Sincerus caffer* and *Hippopotamus amphibibius*, and globally threatened species of mammals like *Panthera leo* and *Panthra pardus*, which together indicate the area for biodiversity conservation. Twenty seven mammalian species were identified in the study area. Order Artiodactyla had the highest number of species (11 species), followed by the orders Carnivora (9 species). While, orders Rodentia and Tubulidentata were represented each by one species. Mammal species distribution and abundance in study area vary due to vegetation types and resource availability. Riparian forest had the highest number of species, followed by Savanna grassland. The distribution and utilization of different vegetation communities by mammal species could be explained in terms of seasonal changes. The habitat associations of the medium and large-sized mammalian species were influenced by seasonal variations in the quality and abundance of forage.

Effective conservation and management strategies are necessary to ensure the sustainable conservation of medium and large mammals in the DWS. For instance, wildlife habitats in the DWS must be linked to the Arjo-Diga protected forest in the northern part [47] and the Haro Abba Diko Controlled hunting area in the west using landscape connectivity. If proper conservation and mitigation measures are not put in place, the study area's large mammals, including *Syncerus caffer*, *Panthera leo*, *Panthera paradus*, and *Hippopotamus amphibius*, will perish. However, the increasing of both small and large-scale farming operations is the main factor endangering the survival of mammals in the study areas [24,48], and other investments should not be considered inside the study area's possible animal habitats. Furthermore, future conservation and management efforts should focus on resolving the issue of human activities such habitat fragmentation by creating adequate corridors inside the research area to reunite the separated areas of good wildlife habitat[49] (S2 Fig in S1 Appendix). Generally, the federal and regional governments should reconsider DWS's current state and make plans to improve local wildlife and habitat conservation.

## Supporting information

**S1 Appendix. Supplementary.**
(DOCX)

## Acknowledgments

We thank to IDEA-WILD for their material support. The authors would like to acknowledge Arjo Dhidhessa Sugar Factory, local communities and district experts who assisted the field survey. Also, they would like to thank Ashetu Kejela for preparing the map of the study area.

## Author contributions

**Conceptualization:** Girma Gizachew Tefera, Tadesse Habtamu Tessema, Tibebu Alemu Bekere, Tariku Mekonnen Gutema.

**Data curation:** Girma Gizachew Tefera.

**Formal analysis:** Girma Gizachew Tefera, Tariku Mekonnen Gutema.

**Investigation:** Girma Gizachew Tefera, Tadesse Habtamu Tessema, Tariku Mekonnen Gutema.

**Methodology:** Girma Gizachew Tefera, Tadesse Habtamu Tessema, Tibebu Alemu Bekere, Tariku Mekonnen Gutema.

**Resources:** Girma Gizachew Tefera, Tadesse Habtamu Tessema.

**Supervision:** Tadesse Habtamu Tessema, Tibebu Alemu Bekere, Tariku Mekonnen Gutema.

**Validation:** Tadesse Habtamu Tessema, Tibebu Alemu Bekere, Tariku Mekonnen Gutema.

**Visualization:** Girma Gizachew Tefera, Tadesse Habtamu Tessema, Tibebu Alemu Bekere, Tariku Mekonnen Gutema.

**Writing – original draft:** Girma Gizachew Tefera.

**Writing – review & editing:** Girma Gizachew Tefera, Tadesse Habtamu Tessema, Tibebu Alemu Bekere, Tariku Mekonnen Gutema.

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
