## [Decision Letter · Decision Letter 0]

15 Oct 2024

PONE-D-24-30486The diversity and habitat association of medium and large mammals  in the Dhidhessa Wildlife Sanctuary, Southwestern EthiopiaPLOS ONE

Dear Dr. Tefera,

Thank you for submitting your manuscript to PLOS ONE. After careful consideration, we feel that it has merit but does not fully meet PLOS ONE’s publication criteria as it currently stands. Therefore, we invite you to submit a revised version of the manuscript that addresses the points raised during the review process.

We look forward to receiving your revised manuscript.

Kind regards,

Tunira Bhadauria, Ph.D.

Academic Editor

PLOS ONE

Journal Requirements:

2. We note that your Data Availability Statement is currently as follows: “All relevant data are within the manuscript and in Supporting Information files.”

Please confirm at this time whether or not your submission contains all raw data required to replicate the results of your study. Authors must share the “minimal data set” for their submission. PLOS defines the minimal data set to consist of the data required to replicate all study findings reported in the article, as well as related metadata and methods (https://journals.plos.org/plosone/s/data-availability#loc-minimal-data-set-definition). For example, authors should submit the following data: - The values behind the means, standard deviations and other measures reported; - The values used to build graphs; - The points extracted from images for analysis. Authors do not need to submit their entire data set if only a portion of the data was used in the reported study. If your submission does not contain these data, please either upload them as Supporting Information files or deposit them to a stable, public repository and provide us with the relevant URLs, DOIs, or accession numbers. For a list of recommended repositories, please see https://journals.plos.org/plosone/s/recommended-repositories. If there are ethical or legal restrictions on sharing a de-identified data set, please explain them in detail (e.g., data contain potentially sensitive information, data are owned by a third-party organization, etc.) and who has imposed them (e.g., an ethics committee). Please also provide contact information for a data access committee, ethics committee, or other institutional body to which data requests may be sent. If data are owned by a third party, please indicate how others may request data access.

4. We note that Figure 1 in your submission contain map/satellite images which may be copyrighted. All PLOS content is published under the Creative Commons Attribution License (CC BY 4.0), which means that the manuscript, images, and Supporting Information files will be freely available online, and any third party is permitted to access, download, copy, distribute, and use these materials in any way, even commercially, with proper attribution. For these reasons, we cannot publish previously copyrighted maps or satellite images created using proprietary data, such as Google software (Google Maps, Street View, and Earth). For more information, see our copyright guidelines: http://journals.plos.org/plosone/s/licenses-and-copyright. We require you to either (a) present written permission from the copyright holder to publish these figures specifically under the CC BY 4.0 license, or (b) remove the figures from your submission:

a. You may seek permission from the original copyright holder of Figure 1 to publish the content specifically under the CC BY 4.0 license. We recommend that you contact the original copyright holder with the Content Permission Form (http://journals.plos.org/plosone/s/file?id=7c09/content-permission-form.pdf) and the following text: “I request permission for the open-access journal PLOS ONE to publish XXX under the Creative Commons Attribution License (CCAL) CC BY 4.0 (http://creativecommons.org/licenses/by/4.0/). Please be aware that this license allows unrestricted use and distribution, even commercially, by third parties. Please reply and provide explicit written permission to publish XXX under a CC BY license and complete the attached form.” Please upload the completed Content Permission Form or other proof of granted permissions as an "Other" file with your submission. In the figure caption of the copyrighted figure, please include the following text: “Reprinted from [ref] under a CC BY license, with permission from [name of publisher], original copyright [original copyright year].”

 b. If you are unable to obtain permission from the original copyright holder to publish these figures under the CC BY 4.0 license or if the copyright holder’s requirements are incompatible with the CC BY 4.0 license, please either i) remove the figure or ii) supply a replacement figure that complies with the CC BY 4.0 license. Please check copyright information on all replacement figures and update the figure caption with source information. If applicable, please specify in the figure caption text when a figure is similar but not identical to the original image and is therefore for illustrative purposes only. The following resources for replacing copyrighted map figures may be helpful: USGS National Map Viewer (public domain): http://viewer.nationalmap.gov/viewer/ The Gateway to Astronaut Photography of Earth (public domain): http://eol.jsc.nasa.gov/sseop/clickmap/ Maps at the CIA (public domain): https://www.cia.gov/library/publications/the-world-factbook/index.html and https://www.cia.gov/library/publications/cia-maps-publications/index.html NASA Earth Observatory (public domain): http://earthobservatory.nasa.gov/ Landsat: http://landsat.visibleearth.nasa.gov/ USGS EROS (Earth Resources Observatory and Science (EROS) Center) (public domain): http://eros.usgs.gov/# Natural Earth (public domain): http://www.naturalearthdata.com/

Reviewers' comments:

Reviewer's Responses to Questions

**Comments to the Author**

1. Is the manuscript technically sound, and do the data support the conclusions?

Reviewer #1: Yes

Reviewer #2: Partly

2. Has the statistical analysis been performed appropriately and rigorously? 

Reviewer #1: Yes

Reviewer #2: No

3. Have the authors made all data underlying the findings in their manuscript fully available?

Reviewer #1: Yes

Reviewer #2: Yes

4. Is the manuscript presented in an intelligible fashion and written in standard English?

Reviewer #1: Yes

Reviewer #2: Yes

5. Review Comments to the Author

Reviewer #1: The paper has well done with some relevant comments shown below:

= Where is a short title in the title page?

= Explain abbreviations in the abstract section.

= Do not repeat words or phrases from the topic directly at the keywords section. Please find equivalent words for them.

= Check and revise the number of mammalian species and their endemics in Ethiopia you mentioned in the introduction section.

= List all the etc in the document.

= How you distributed the 25 transects in different habitat you used?

= The whole area covers about 1300 km2 as you mentioned in the study area description above. How this calculation came? Does this percent of the sampled area represent the whole study area?

= How you avoided double counts of the animals?

= Does the month April the wet season in your study area?

= Why not early morning at 6 am? Why you started from 8 am?

= Show the total transect lines first before the actual transects you used under table 1.

Reviewer #2: The authors present a commendable research concept with the potential to contribute meaningfully to conservation efforts. However, several critical issues need to be addressed before publication.

One significant concern is the lack of data on habitat association and the absence of any ecological theory related to this concept. The authors' interchangeable use of "habitat association" and "habitat occupancy" is misleading, as these terms represent distinct ecological phenomena. Without supporting data, it is advisable to reconsider the inclusion of habitat association throughout the work.

Moreover, the authors' acknowledgment of human interference (sugarcane plantation) and four distinct habitat types, coupled with seasonal data collection, necessitates a rigorous statistical analysis. The application of Generalized Linear Models (GLM) followed by model selection would provide a robust framework for understanding how these factors influence wildlife diversity and abundance.

I have attached a detailed annotated PDF outlining specific comments and suggestions for improvement. It is crucial that the authors carefully review this feedback and make the necessary revisions to enhance the paper's clarity, rigor, and overall contribution to the field.

Regarding the submission of two identical manuscripts, I recommend that the authors carefully review the entire document and address the comments provided in the annotated PDF. This will ensure that the final version is well-structured, coherent, and professionally presented.

6. PLOS authors have the option to publish the peer review history of their article (what does this mean? ). If published, this will include your full peer review and any attached files.

**Do you want your identity to be public for this peer review?** For information about this choice, including consent withdrawal, please see our Privacy Policy .

Reviewer #1: No

Reviewer #2: No

---

## [Author Response · Author response to Decision Letter 1]

9 Dec 2024

Date; - November, 7, 2024

To: Tunira Bhadauria, Ph.D.

Academic Editor

Journal of PLOS ONE

Subject: Responses to the reviewers’ comments

Dear Dr. Stephanie S. Romanach,

We are very grateful for your and reviewers’ critical and constructive comments regarding our manuscript numbered PONE-D-24-30486, entitled “The diversity and habitat association of medium and large mammals in the Dhidhessa Wildlife Sanctuary, Southwestern Ethiopia”. Please find re-submitted (the revised version) of our manuscript in which we have carefully considered and addressed all comments raised by the referees. The revised version of the manuscript includes many improvements and summarized below. Moreover, the revised manuscript is presented using track-change with red color. We hope our responses are satisfactory and you will find this version suitable for publication in Journal of PLOS ONE. We look forward to hearing from you.

Dear Academic Editor, please note the following points

1. Our responses to the reviewers’ comments are provided

2. The line numbers with red colors (line#) refer to the improved version of the revised manuscript

Best regards,

Corresponding author

Response to Academic editor #1

1. Comment 1(General comment), the reviewers suggested that the manuscript when submitting your revision, we need you to address these additional requirements.

• The manuscript should meet PLOS One's style requirements.

• Data availability by the authors upon reasonable request.

• Figures 1 in your submission contain [map/satellite] images, which may be copyrighted.

Response: Thank you very much for your suggestion. We agree with the comment and we made improvements to the issues mentioned above. However, we have already referenced Figure 1 on Line# 130.

2. Comment 2, Line 1: The reviewer one suggested that; - where is a short title in the title page? Explain abbreviations in the abstract section; do not repeat words or phrases from the topic directly at the keywords section. Please find equivalent words for them.

Response: Thank you very much for your suggestion. We agree with the comment and made correction. Based on the comments and suggestions using a track change from Line# 32and 52 and we have re-written or revised the whole of portions the abstract. However, the guidelines of PLOS ONE stated that “short title” should be written in the submission system.

3. Comment 3, Line 1: The lack of data on habitat association and the absence of any ecological theory related to this concept.

Response: Thank you very much for your comments and suggestions. We have made correction as commented on Line# 74-82.

4. Comment 4, Line 59: “Check and revise the number of mammalian species and their endemics in Ethiopia you mentioned in the introduction section“

Response: Thank you very much for your suggestions. We have made correction as commented on Line# 60.

5. Comment 5, Line 62: Change the second paragraph into third paragraph and provide detail evidence on threats and level of impacts from global to local

Response: Thank you very much for your comments and suggestions. We have changed and revised as commented on Line# 83-108.

6. Comment 6, Line 102: what is the objective of the study?

Response: Thank you very much for your suggestions. We agree with the comment and made correction by rewriting the words using a track change from Line# 121-123

7. Comment 7, Line 104-120: Detail information of the study area like map of the study area, historical background, loss of the area, purpose of establishment, and who is responsible to manage?

Response: Response: Thank you very much for your comments. We agree with the comment and made correction based on our objectives of the study from Line# 125-153

8. Comment 8, Line 131: List all the “etc” in the document

Response: Thank you very much for your suggestions. We agree with the comment and made correction by rewriting the words using a track change from Line# 155-161

9. Comment 9, Line 134 and 135: Incomplete idea and make it on new paragraph

Response: Thank you very much for your suggestions. We have made correction on line #160 and 162

10. Comment 10, Line 146-160: Avoid repetition, clarity, how do they understand transect width to count spp. If not experts and uses separate sentence to explain how each indirect methods are used in your study.

Responses: Thank you very much for your comments. We agree with the comment and made correction., Based on the comments and suggestions using a track change from Line# 164-184, we have re-written or revised the whole of those portions.

11. Comment 11, Line 164: How you distributed the 25 transects in different habitat you used?

Response: Thank you very much for your comments/suggestions. We have used based on area cover or size of the habitat (line #515).

12. Comment 12, Line 165: The whole area covers about 1300 km2 as you mentioned in the study area description above. How this calculation came? Does this percent of the sampled area represent the whole study area?

Response: Thank you very much for your suggestions. The Ethiopian government formerly designed the Dhidhessa Wildlife Sanctuary in (line #138)1970 with the area 1300 km2 (line #128). However, at the moment, the area was used for various purposes. Based on these, we selected a sample from the whole 98.56 km2 (line #515) of the area; therefore, the 25% represents the sampled area rather than the whole area

13. Comment 13, line 169: “How you avoided double counts of the animals?

Response: Thank you very much for your comments. We agree with the comment and made correction from Line# 171-173 “As you indicated, in order to prevent double counting, adjacent transect lines were distributed one to two kilometers apart”.

14. Comment 14, Line 170: Does the month April the wet season in your study area?

Why not early morning at 6 am? Why you started from 8 am?

Response: Thank you very much for your comments and suggestions. We agree with the comment and made correction using a track change from Line# 172-173.

15. Comment 15, Line 515: Show the total transects lines first before the actual transects you used under table 1.

Response: Thank you very much for your comments. We agree with the comment and made correction using a track change from Line# 515.

16. Comment 16, Line 162-171: For analysis method use software

Response: Thank you very much for your comments and suggestions. We agree with the comment and made correction using a track change from Line# 186-199 and 225 (as indicated that, we use PAST4 project software for the analysis of our study)

17. Comment 17, Line 208: “Not method provided”

Response: Thank you very much for your suggestions. We agree with the comment and made correction for Line# 232 with track changes.

18. Comment 18, Line 221: The discussion section should follow logical flow of idea as it was presented in result section.

Response: Thank you very much for your comments and suggestions. We agree with the comment and made correction using a track change from Line# 245, 292,310 and 320 (this section was re-written, as suggested).

19. Comment 19, Line 238,240,249: How do you compare Sanctuary with National Park?

Response: Thank you very much for your comments. However, we made the comparisons based on the methodologies and objectives rather than the area's size from Line# 249.

20. Comment 20, Line 267,271: Out of your study context, you didn’t use this method to collect data, as evidence lacks in method section.

Response: Thank you very much for your comment. We agree with the comment and made correction using a track change from Line# 298-303 ("involving" re-write and re-worded as suggested).

21. Comment 21, Line 296: Include implication of the finding, not only summarizing, what is the specific recommendation based on your finding.

Response: Thank you very much for your comment. We agree with the comment and made correction using a track change from Line# 330.

22. Comment 22, Line 328: Do you think this contribution only qualifies them for authorship? It’s very important to consider all valuable contributions.

Response: Thank you very much for your comments and suggestions. We agree with the comment and made correction using a track change from Line# 365-369.

23. Comment 23, Line 482: change “occurrence”

Response: Thank you very much for your suggestions. We have changed as commented, on #Line 531.

24. Comment 24: Use high resolution map of study area-300dpi.

Response: Thank you very much for your comment. We agree with the comment and made correction.

---

## [Editor Report · Decision Letter 1]

30 Dec 2024

The diversity and habitat association of medium and large mammals  in the Dhidhessa Wildlife Sanctuary, Southwestern Ethiopia

PONE-D-24-30486R1

Dear Dr. Tefera 

We’re pleased to inform you that your manuscript has been judged scientifically suitable for publication and will be formally accepted for publication once it meets all outstanding technical requirements.

Kind regards,

Tunira Bhadauria, Ph.D.

Academic Editor

PLOS ONE
---

## [Editor Report · Acceptance letter]

PONE-D-24-30486R1

PLOS ONE

Dear Dr. Tefera,

I'm pleased to inform you that your manuscript has been deemed suitable for publication in PLOS ONE. Congratulations! Your manuscript is now being handed over to our production team.

Kind regards,

on behalf of

Dr. Tunira Bhadauria

Academic Editor

PLOS ONE